# Impaired Nitric Oxide Synthetase Activity in Primary Ciliary Dyskinesia—Data-Driven Hypothesis

**DOI:** 10.3390/jcm12186010

**Published:** 2023-09-16

**Authors:** Lisa Eggenkemper, Anne Schlegtendal, Christoph Maier, Thomas Lücke, Folke Brinkmann, Bibiana Beckmann, Dimitrios Tsikas, Cordula Koerner-Rettberg

**Affiliations:** 1University Children’s Hospital, Ruhr-University Bochum, 44791 Bochum, Germany; anne.schlegtendal@rub.de (A.S.); christoph.maier@rub.de (C.M.); thomas.luecke@rub.de (T.L.); folke.brinkmann@uksh.de (F.B.); cordula.koerner-rettberg@prohomine.de (C.K.-R.); 2Department of Internal Medicine and Gastroenterology, Christophorus-Kliniken Coesfeld, Teaching Hospital of University Münster, 48653 Coesfeld, Germany; 3Section for Pediatric Pneumology and Allergology, University Medical Center Schleswig-Holstein, 23538 Lübeck, Germany; 4Core Unit Proteomics, Institute of Toxicology, Hannover Medical School, Carl-Neuberg-Str. 1, 30625 Hannover, Germany; beckmann.bibiana@mh-hannover.de (B.B.); tsikas.dimitros@mh-hannover.de (D.T.); 5Department of Pediatrics, Marien-Hospital Wesel, Teaching Hospital of University of Münster, 46483 Wesel, Germany

**Keywords:** primary ciliary dyskinesia, nitric oxide, nitric oxide synthetase, cystic fibrosis, L-arginine

## Abstract

Low nasal nitric oxide (nNO) is a typical feature of Primary Ciliary Dyskinesia (PCD). nNO is part of the PCD diagnostic algorithm due to its discriminative power against other lung diseases, such as cystic fibrosis (CF). However, the underlying pathomechanisms are elusive. To better understand NO dysregulation in PCD, the L-arginine/NO (Arg/NO) pathway in patients with PCD (pwPCD) and CF (pwCF) and in healthy control (HC) subjects was investigated. In a prospective, controlled study, we measured in 24 pwPCD, 25 age-matched pwCF, and 14 HC the concentrations of the NO precursors Arg and homoarginine (hArg), the arginase metabolite ornithine (Orn), the NO inhibitor asymmetric dimethylarginine (ADMA), and the major NO metabolites (nitrate, nitrite) in sputum, plasma, and urine using validated methods. In comparison to HC, the sputum contents (in µmol/mg) of L-Arg (PCD 18.43 vs. CF 329.46 vs. HC 9.86, *p* < 0.001) and of ADMA (PCD 0.055 vs. CF 0.015 vs. HC 0.010, *p* < 0.001) were higher. In contrast, the sputum contents (in µmol/mg) of nitrate and nitrite were lower in PCD compared to HC (nitrite 4.54 vs. 9.26, *p* = 0.023; nitrate 12.86 vs. 40.33, *p* = 0.008), but higher in CF (nitrite 16.28, *p* < 0.001; nitrate 56.83, *p* = 0.002). The metabolite concentrations in urine and plasma were similar in all groups. The results of our study indicate that PCD, unlike CF, is associated with impaired NO synthesis in the lung, presumably due to mechano-chemical uncoupling.

## 1. Introduction

Primary ciliary dyskinesia (PCD) is a rare, autosomal recessively inherited disease mainly associated with respiratory symptoms. In PCD, various genetic defects of the motile respiratory cilia lead to an impairment in ciliary function, which, in turn, compromises mucociliary clearance [1]. Sequelae caused by reduced mucociliary clearance include upper and lower airway infections, eventually resulting in progressive bronchiectasis and impairment of lung function. The diagnostic algorithm includes high frequency video microscopy (HFVM), transmission electron microscopy (TEM), nasal nitric oxide (nNO) measurement, and genetic analysis [2,3,4]. Low nNO levels are characteristic of PCD and are of diagnostic value [5]. However, the pathogenesis of low nNO levels as a hallmark of PCD is still unclear.

Nitric oxide (NO), originally known as endothelium-derived relaxing factor (EDRF), is a small, gaseous free radical with many physiological homeostatic and immunological functions [6]. The half-life of NO in blood is less than 0.1 s. NO is oxidized to nitrite and nitrate, which are useful measures of NO synthesis. NO is produced by C- and N-oxidation of the guanidino group of L-arginine (Arg) catalyzed by the nitric oxide synthase (NOS) family, with L-citrulline (Cit) being the second product of this reaction. At least three NOS isoforms are known: the constitutive and Ca^2+^-dependent neuronal NOS (nNOS), endothelial NOS (eNOS), and the inducible Ca^2+^-independent NOS (iNOS) [6,7]. iNOS is expressed in cells activated by cytokines and lipopolysaccharides (LPS) and acts via a high-affinity calmodulin-binding site [6,8]. iNOS is expressed across a wide range of tissues; co-expression of iNOS and nNOS with differential regulation has been demonstrated in lung epithelial cells [8]. nNOS is expressed in many different cell types such as skeletal, cardiac, and smooth muscle cells, pancreatic islet cells, macula densa cells of the kidney, as well as in alveolar and bronchial epithelial cells [8]. Recently, nNOS has been localized in murine ciliated tracheal epithelial cells in close proximity to the apical cortical actin grid, basal bodies, and in the ciliary axonemes and was linked to physiological ciliary function [9,10]. 

The activity of all NOS isoforms is mainly inhibited by asymmetric dimethylarginine (ADMA) [7]. ADMA is formed by post-translational modification, i.e., asymmetric dimethylation of Arg residues in proteins. ADMA is metabolized by dimethylarginine dimethylaminohydrolase (DDAH) to Cit and dimethylamine (DMA) [11]. High ADMA levels were found in many cardiovascular (e.g., heart failure and pulmonary hypertonia) and pulmonary diseases (e.g., bronchial asthma and cystic fibrosis), and in renal failure [11,12].

Cystic fibrosis (CF) is an autosomal recessive disease caused by mutations in the cystic fibrosis transmembrane conductance regulator (CFTR) gene coding for the chloride channel situated in the apical membrane of epithelial cells of exocrine tissue. CFTR dysfunction results in viscous mucus impairing mucociliary clearance [13]. Patients with CF (pwCF) also show low nNO levels, lying between those of patients with PCD (pwPCD) and healthy humans [14], as well as an upregulated Arg/NO pathway as evidenced by increased plasma NO metabolites [15]. Higher ADMA concentrations were found in the sputum of pwCF compared to healthy humans [16]. ADMA concentrations decreased after treatment of pulmonary exacerbations, whereas sputum NO metabolites and fractionally exhaled NO (FENO) increased [17]. High levels of ADMA are suspected to lead to bronchial obstruction via NOS inhibition, leading to lower NO production [16,17]. Higher Arg, ADMA, nitrite, and nitrate concentrations in plasma and nitrate in urine were measured in pediatric pwCF [15]. 

Compared to the considerable amount of data on the Arg/NO pathway in CF, such data is lacking in PCD. Few hypotheses deal with the lower nNO levels in PCD. They include impaired NO biosynthesis, elevated NO oxidation to nitrite and nitrate, or NO trapping in obstructed paranasal sinuses [18]. A positive correlation between nNO levels and expression of the iNOS gene in pwPCD was reported [19]. A correlation between low nNO levels and increased quantity of immotile cilia has also been observed [19]. In vitro, the ciliary beat frequency (CBF) was elevated by treatment with Arg of human paranasal sinus mucosa, which expresses iNOS and eNOS, indirectly indicating an effect of NO synthesis on CBF [20,21]. However, a firm attribution of low nNO levels to impaired activity of a specific NOS isoform has not been proven yet. Recently, the nNOS isoform has been linked to ciliary function [9]. 

The primary aim of this study was to analyze whether the L-Arg/NO pathway is different in PCD, CF, and healthy controls (HC). We hypothesized that pwPCD and pwCF show higher sputum concentrations of Arg and ADMA in comparison to HC. However, we hypothesized that the Arg/NO pathway behaves differently in PCD and CF and that these differences account for the PCD-immanent very low nNO levels.

## 2. Materials and Methods

### 2.1. Study Design and Population

We conducted a single-center, prospective, controlled clinical trial to assess the L-Arg/NO pathway in pwPCD in comparison to pwCF and HC. PwPCD and pwCF were recruited within trimestral routine examination in the PCD special outpatient clinic or the CF Centre at University Children’s Hospital Bochum. Between 12/2017 and 05/2020, overall, 34 of 63 pwPCD and 47 of 88 pwCF above the age of ten years registered in the hospital were asked about their interest in study participation. The recruitment process is depicted in Figure 1. For an age-matched HC group, children and adults from outside the clinic were invited. 

Inclusion criteria for pwPCD and pwCF were a confirmed diagnosis, age above ten years, and written consent to study participation. Diagnosis of PCD was verified by HFVM, TEM, IF, and molecular genetics, according to ERS guidelines [3]. For details on the diagnostic parameters, see Appendix A. CF diagnosis was verified by pathological sweat test and molecular genetics. Inclusion criteria for HC were the absence of any acute infection or chronic lung disease, age above ten years, and written consent for study participation. Study participants were excluded when canned fish was consumed the day before and if a special diet was followed in the last two weeks before the examination date. We also excluded patients with colonization with resistant strains due to hygienic restrictions and with the inability to produce sputum. As part of the routine examinations, body weight and length were measured. Oral and inhaled antibiotic therapy over three months was classified as long-term antibiotic therapy. 

The local ethics committee of the Ruhr-University Bochum, Germany, approved the study in 2017 (Ethics Committee No. 17-6079 dated 3 September 2017).

### 2.2. Biochemical Analysis in Sputum, Plasma, and Urine

After a fasting period not less than four hours, 5 mL venous blood was sampled using Lithium–Heparin Monovettes (Kabe Labortechnik, Nümbrecht-Elsenroth, Germany), brought to the laboratory on ice and centrifuged immediately (2000× *g*, 10 min, 4 °C) using a Megafuge 1.0R from Heraeus (Stadt, Germany). Urine was collected by spontaneous micturition. At least 1 mL aliquots of plasma and urine were stored at −80 °C until further analysis. Sputum samples were immediately quick-frozen in liquid nitrogen and stored at −80 °C until further analysis. Lung-healthy individuals do not produce sputum; therefore, inhaling hypertonic saline under constant control of the maximum flow rate using a peak flow meter according to ERS guidelines [22] induced sputum in HC. From pwPCD and pwCF, additional blood samples were taken to measure C-reactive protein (CRP) and leukocytes and sputum for culture-based microbiology. 

Plasma, urine, and sputum samples were analyzed using high-performance liquid chromatography (HPLC) for ornithine (Orn) and citrulline (Cit) and with gas chromatography–mass spectrometry (GC-MS) for all other parameters, as described previously [23,24,25]. The sputum parameters are reported as µmol analyte per mg sputum. Analyte concentrations in urine were corrected for urinary creatinine excretion and are reported as µmol analyte per mmol creatinine. All analyses for the Arg/NO pathway were performed at the same time in one laboratory.

### 2.3. Lung Function

The values of nNO and FENO were assessed with EcoMedics CLD 88sp (Duernten, Schweiz) according to current recommendations [26]. Both values were measured in parts per billion (ppb) and converted into nanoliters per minute (nL/min). Spirometry was performed using the MasterScreen Body/Diff (Vyaire, Hoechberg, Germany) according to ATS/ERS guidelines [27]. Multiple breath washout was measured with the Exhalyzer D (EcoMedics AG, Duernten, Switzerland) according to current guidelines [28]. We reported the Lung Clearance Index (LCI) of 2.5, as recommended [29]. All pulmonary function tests were done by specially trained staff.

### 2.4. Statistical Analyses and Sample Size

As there are no previous values for Arg/NO metabolism in pwPCD, we assumed that a difference of a factor of 2 in the normal values for arginase and ADMA in sputum is clinically relevant. Using 90% power (alpha = 0.05), the sample size calculation resulted in a ratio of pwPCD and the HC of 24:12. Statistical analyses were performed using the statistical software package IBM^®^ SPSS^®^ Statistics Version 29.01.0 for Mac OS (IBM Corp., Armonk, NY, USA). The equality of variances was tested with the Levene test. When the variance of mean values was equal, the significance (*p* < 0.05) was tested with ANOVA. In cases of unequal variance of mean values, the significance (*p* < 0.05) was tested using the Welch test. Post-hoc analysis was performed using Bonferroni (equal variance of mean values) or Dunnett-T3 (unequal variance of mean values). Data are presented as mean ± standard deviation (SD) or median (25–75th interquartile range). Missing data were considered in the analyses and are marked separately in the results. 

## 3. Results

### 3.1. Characterization of Participants, Bacterial Colonization, and Lung Function 

A total of 24 pwPCD (age 9.7–31.6 years), 25 pwCF (age 11.0–34.7 years), and 14 HC (age 10.1–33.3 years) were enrolled. Half of the pwPCD had a situs inversus, consistent with PCD features. Pancreatic insufficiency was present in 92% of the pwCF. Three (12.5%) pwPCD and one (4%) pwCF were overweight (BMI z-Score > 1.96). Nine (36%) pwCF received CFTR modulator treatment before enrolment into the study (see Appendix A). Apart from the CFTR modulators, pwPCD and pwCF received similar supportive therapy (secretolysis, antibiotics) according to therapy standards. Demographic and functional data of patients and controls are shown in Table 1. Pathological airway colonization was more frequently detectable in pwCF, especially with those infected by *Pseudomonas aeruginosa* (Appendix A). This may explain the higher number of long-term antibiotic treatments (Table 1). nNO levels were significantly lower in pwPCD compared to pwCF and HC (Table 1, Figure 2A). Subgroup analysis of pwPCD showing immotile cilia (*n* = 16) vs. dyskinetic cilia (*n* = 5) in HFVM did not show a systemic difference in nasal NO value (mean value 11.57 nL/min vs. 9.88 nL/min). Lung function between patients of both lung diseases did not differ significantly (Table 1). 

### 3.2. Characterization of the L-Arg/NO Pathway in the Sputum

The L-Arg/NO pathway was found to be altered in PCD and CF compared to HC (Table 2). In sputum, the concentrations of Arg, hArg, and ADMA were significantly higher in the patient groups, being more pronounced in CF than in PCD. The sputum concentration of ADMA was 5.5 times higher in PCD and 15 times higher in CF than in HC. The Arg/ADMA ratio, a measure of NO biosynthesis capacity, was significantly lower in both diseases. The Orn/Cit ratio, a measure of arginase activity, was markedly higher in both lung diseases. This ratio was higher in CF (14.3-fold) than in PCD (6.4-fold). In the sputum, the Orn/Cit ratio correlated with the Arg concentration in all three groups. The highest Pearson correlation was observed in PCD (R 0.94; Appendix A). 

Discordant results were observed with respect to nitrite and nitrate sputum concentrations. They were significantly lower in pwPCD but higher in pwCF compared to HC (Figure 2, Table 2). Figure 3 visualizes the state of the Arg/NO pathway in sputum in pwPCD and pwCF compared to HC schematically.

### 3.3. Characterization of the L-Arg/NO Pathway in Plasma and Urine

In plasma and urine (Appendix A), less significant differences between these three groups were observed. In PCD and CF, slightly higher plasma concentrations of nitrite were measured. In pwPCD, plasma nitrate and urinary nitrate excretion (*p* = 0.006) were lower compared to healthy controls, whereas in CF, urinary nitrate excretion was higher. The plasma Orn/Cit ratio was higher in CF but comparable in PCD and HC. Only in CF disease a higher concentration of L-Arg (however, not reaching significance) was measured in plasma and urine.

## 4. Discussion

To our knowledge, this is the first study to analyze the Arg/NO pathway in pwPCD. In comparison to HC and pwCF, considerable differences in the Arg/NO pathway were observed in the sputum of the pwPCD. Higher sputum levels of ADMA, Arg, and the Orn/Cit ratio were found in both PCD and CF. However, nasal NO levels were lower in PCD compared to CF and HC, and the concentrations of the NO metabolites nitrite and nitrate were lower in pwPCD while being higher in the sputum of pwCF when compared to HC. These results point to different states of the Arg/NO pathway in health and disease and clearly discriminate the Arg/NO pathway in the lung compartment in PCD from CF disease.

In pwPCD, NO tends to be low not only in the sinonasal but also in the bronchial compartment [18], but in the latter compartment lacking discriminative power. PCD immanent alterations in NO synthesis or metabolism will probably affect both sinonasal and bronchial mucosae, which both bear cilia. Therefore, measuring NO metabolites in sputum gives potentially relevant data, as sputum is a compartment in direct proximity to bronchial mucosa. 

In plasma and urine, differences in the Arg/NO pathway were discrete or not detectable in the three groups, suggesting no major differences in the systemic and whole-body NO synthesis, respectively. Thus, the differences found in the sputum of the subjects of the groups are indicative of lung-specific alterations in PCD and CF. 

### 4.1. Similarities between PCD and CF

The hypothesis of an altered L-Arg/NO pathway in the pulmonary compartment in PCD was motivated by the facts that PCD and CF diseases share a profound neutrophilic inflammatory airway pathology and that in CF, L-Arg/NO pathway alterations linked to the CF airway inflammation have already been described. In CF, low exhaled airway NO [14,31,32,33] goes along with increased sputum arginase activity [33], increased levels of the endogenous NOS inhibitor ADMA [16], reduced L-Arg/ADMA ratio [16], and increased polyamines [34]. In our study, we found higher L-Arg, hArg, and ADMA levels, a lower L-Arg/ADMA ratio, and a higher Orn/Cit ratio in the lung compartment of our CF cohort, being in line with previous data and expanding them. For the first time, we show these alterations also in PCD disease, paralleling those in CF.

The higher Orn/Cit ratio in sputum of both pwCF and pwPCD suggests a shift in the balance between arginase and NOS activity towards Orn production, possibly mediated or augmented through inhibition of the NOS by high ADMA levels, as ADMA is an endogenous NOS inhibitor. Hence, it can be speculated that a reduced bioavailability of the substrate L-Arg for the NOS through NOS inhibition by ADMA and an increased activity of the competing arginase enzymes shown in CF [33,35] also may play a role in PCD due to similar airway inflammatory states. In our study, differences in L-Arg and ADMA levels were significantly more pronounced in sputum from the pwCF than from the pwPCD, which could be attributed to more pronounced airway inflammation in CF. Airway inflammation was not measured in our study, but bacterial colonization was significantly higher in the CF cohort compared to the PCD cohort, and only a minority of pwCF were treated with CFTR modulators at the time of study conductance (Appendix A). 

All in all, elevated concentrations of ADMA, L-Arg, an increased Orn/Cit ratio, and decreased L-Arg/ADMA ratio are present in both lung diseases PCD and CF and, thus, are not disease-specific but might rather be a function of airway inflammation shared by both diseases. 

### 4.2. PCD-Specific Effects

Previous studies reported higher concentrations of the NO metabolites nitrite and nitrate in CF sputum compared to healthy controls [36] despite low exhaled NO in CF [14,27,29,33]. This is believed to be due to an increased NO metabolism in the CF sputum by NO retention in the airway surface liquid compartment and/or degradation by denitrifying bacteria [37] and phagocytes. In the present study, we found significantly higher nitrite and nitrate concentrations in CF sputum compared to sputum from HC, consistent with previous data [34]. Remarkably, in the present study, PCD sputum nitrite and nitrate levels were lower compared to HC and CF, strongly discriminating PCD L-Arg/NO metabolism from that in CF. This marked reduction in NO downstream metabolites confined to the PCD cohort parallels their distinctly low nasal NO, being even lower than in CF (Table 1), which is in line with nasal NO known to be specifically low in PCD [35] and, thus, being used in PCD diagnostic algorithms [26,38]. Low NO downstream metabolites in PCD sputum but not in CF sputum suggest that the PCD-specifically low nasal NO levels result from a reduction in NO production, i.e., reduced NOS activity, due to mechanism(s) not shared between PCD and CF. 

Several hypotheses about NO dysregulation in PCD have been proposed in the literature and comprehensively discussed by Walker et al. [18]. However, the underlying mechanism still has not been fully elucidated. One hypothesis for decreased nasal NO in PCD is a local inhibition of NOS activity by ADMA. Although we found significantly increased ADMA levels in PCD sputum compared to HC, the amount of ADMA in CF sputum was even higher than in PCD sputum. Therefore, NOS inhibition by ADMA cannot account for the severely low nNO levels in PCD. Furthermore, a substrate deficiency for the NOS was discussed [18], but based on our data, this can be excluded as a reason for decreased NO production in PCD, as L-Arg concentration in PCD sputum is significantly higher than in HC. Increased NO consumption by NO-consuming/denitrifying bacteria and/or by sputum phagocytes as the cause of the very low nasal NO in PCD also seems unlikely, as in our study, pwCF showed significantly more bacterial colonization, higher nasal NO, and higher sputum NO downstream metabolites than pwPCD. The hypothesis that pathology of the PCD upper airways could result in low nNO [18] by gas trapping or reduced NO production capacity confined to the paranasal sinuses is not supported by our measurements: NO was low in both upper and lower airways in PCD, and NO metabolites were low in sputum, hinting at a rather global mechanism not confined to the upper airways. 

The results of the present study suggest that a specific pathomechanism is active in PCD but not in CF and accounts for the reduced airway NO production in PCD, leading to the PCD-immanently low nasal NO. Our data preclude low NO production in PCD being attributable to NOS inhibition by ADMA or to bronchial inflammation, as these mechanisms would also work in pwCF. 

As discussed elsewhere [18,39], low NOS activity in PCD could potentially result from mechanochemical uncoupling. This would imply that normal NOS activity requires ciliary function [21,40]. A “mechano-chemical decoupling” resulting in low NO production by a decreased NOS activity due to reduced NOS mechanical “loading” was discussed in children with Duchenne muscular dystrophy (DMD) [41]. In DMD, a dystrophin deficiency in muscle cells due to mutations in the dystrophin gene goes along with reduced intracellular NO production [42]. In the plasma of pediatric DMD patients, lower nitrate concentrations were measured compared to HC [43]. In muscle cells, the neuronal NOS (nNOS) is anchored to sarcolemma by a dystrophin–glycoprotein complex. In DMD, the missing dystrophin results in an aberrant translocation and accumulation of nNOS in the cytosol, leading to reduced nNOS enzyme activity [44]. However, a PCD-immanent candidate mechanism able to account for reduced NOS activity in respiratory epithelial cells throughout PCD disease types would not be attributable to one gene. This is because PCD is a very heterogenous genetic disease [1] with PCD-causative mutations in at least 50 different genes. Thus, in PCD, a mechanism depending on a single protein defect is not plausible. A decade ago, it was hypothesized [19] that low nasal NO in PCD might be due to an uncoupling of the contractile process of ciliary beating from the enzyme NOS, leading to failure in NO production. Interestingly, in that study, low nasal NO correlated with reduced ciliary beat frequency and inversely with the number of immotile cilia not only in PCD as a genetic disease but also in secondary (i.e., acquired) ciliary defects, linking mechanical ciliary malfunctioning to low NO irrespective of a genetic defect. 

Of note, there is spatial proximity of the NOS, actin, and microtubules in respiratory epithelial cells [9] as well as in muscle cells [44], and the contractile mechanisms of structures at the ciliary base in respiratory epithelial cells and of contractile structures in muscle cells are similar. In an in vitro animal model, mechanically induced muscular action increased neuronal NOS (nNOS) activity, going along with NO formation in muscle cells, a phenomenon called “mechanical loading” [45]. Hence, we postulate that in PCD, the mechanical ciliary malfunction itself, caused by various genetic PCD defects, might be the origin of the markedly low nasal NO via reduced “mechanical loading” of the nNOS (“mechano-chemical decoupling”). This hypothesis is strengthened by the observation of higher nasal NO in pwPCD with residual ciliary function [46] and by very recent data linking nNO production rates to specific PCD genotypes: pwPCD with normal ciliary ultrastructure compared with abnormal ultrastructure showed higher ciliary motility and higher nNO production rates [47], suggesting that higher NO production and higher residual ciliary motility are connected. Unfortunately, due to the low number of cases in our study, a sufficient subgroup analysis of the groups with different genetic mutations or beating phenotypes (like immotile cilia compared to dyskinetic cilia) was not possible. 

A recent work [9] studied the spacial and functional connection of the neuronal NOS to ciliary structures and the impact of NO donors in a knock-out mice model in ciliary beating. In murine respiratory epithelial cells, the nNOS is localized around the docking points of the ciliary bases at the apical cytoskeleton, forming a tight network around the ciliary bases. In nNOS knock-out mice, a lack of neuronal NOS and, thus, of NO production in the respiratory epithelium was linked to a loss of ciliary beating activity, with cilia losing their synchronized beating direction and epithelial cells losing polarity in the absence of NO [9]. After treatment with NO, donors synchronized ciliary function was restored. In summary, this study demonstrated a link between reduced NO production in ciliated epithelial cells and impaired cilia beating function. Thus, an impaired ciliary function might not only compromise NOS activity by mechano-chemical uncoupling but might itself be a secondary consequence of reduced NO production. Similarly, Pifferi et al. [19] speculated that reduced nasal NO levels, as well as being a marker for PCD, may contribute to its pathophysiology. Very recently, in a large PCD cohort, lower nNO observed in patients with more severe genetic and/or ultrastructural PCD types was shown to correlate with greater lung function decline over time and higher nNO with a reduced likelihood of bacterial infection [48]. Taken together, NO production and motile ciliary defects seem to be interrelated in a complex way, forming a vicious circle. Up to now, it can only be speculated whether treatment strategies aiming at increasing NO production (such as pharmacological NO donors) might be able to improve the disease course or prognosis in PCD. 

### 4.3. Strength and Limitations

A strength of the study is the use of reliable GC-MS methods for the quantitative measurement of numerous parameters of the Arg/NO pathway in sputum, plasma, and urine samples of patients and HC. Although groups of patients and HC are relatively small, group effects can clearly be detected. Patients and HC were similar in age distribution. However, due to the limited availability of pwPCD and pwCF, it was not possible to achieve a completely balanced gender distribution. However, gender-specific differences in the Arg/NO pathway have not been described so far. The recruitment period of our study was relatively long, but the biosamples were frozen at −80 °C immediately. All samples were transported to the laboratory together in a frozen state (dry ice) and analyzed at one single time period. A subgroup analysis to correlate between differential genetic, beating phenotype, or ultrastructural defects and the Arg/NO pathway results was not performed due to the small numbers of various genetic backgrounds. A high proportion of pwPCD have genetic defects known to cause low nNO (i.e., dynein arm defects), and only two patients have ciliary defects known to cause either low or normal nNO [46]. However, while this circumstance precludes insights into Arg/NO pathway diversity due to genetic differences, it has been an advantage in terms of recognizing the Arg/NO effects typical for most PCD genotypes. 

## 5. Conclusions

In the present study, the Arg/NO pathway was comprehensively studied for the first time in pwPCD and was compared to pwCF and HC. In sputum, we found marked differences between pwPCD and pwCF and compared to HC. In PCD, lower nitrate and nitrite concentrations in the sputum and lower nasal NO levels suggest decreased NOS activity in the respiratory epithelium in PCD compared to CF and HC. Driven by the presented data, a mechano-chemical uncoupling could be assumed as causative for low nasal NO in PCD because mechanical impairment of motile ciliary function is the hallmark of PCD pathology across many various genetic defects. Dependence of NO production on mechanical loading may have therapeutical implications: In PCD, physical exercise may improve endogenous NO synthesis in the lung by positively affecting the mechanical loading of the NOS in the respiratory epithelium. Pharmacological treatment with NO donors may also be an option. Further studies are needed to confirm the proposed mechano-chemical decoupling of the NOS in PCD and to address its therapeutic augmentation in the clinical course of this disease.

## Figures and Tables

**Figure 1 jcm-12-06010-f001:**
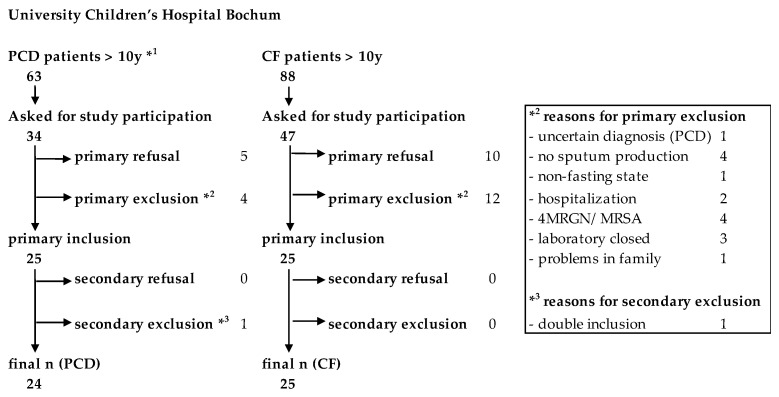
Flow chart of the recruitment process of the study. *^1^Age of one patient below 10 (9.7 years) due to false age information at the time of recruitment. *^3^ One patient mistakenly gave consent twice. Patient data were collected and included only once.

**Figure 2 jcm-12-06010-f002:**
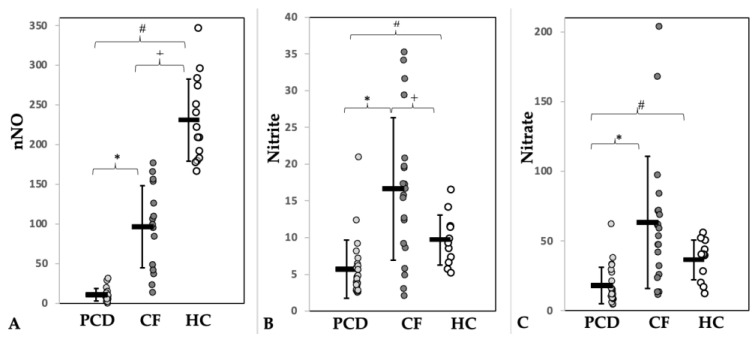
Concentrations of (**A**) nasal nitric oxide (nNO, nL/min) and of its metabolites nitrite (**B**) and nitrate (**C**) in sputum (µmol/mg sputum) of patients with PCD (light grey) or CF (dark grey) and of healthy controls (HC, white). * = Significance PCD vs. CF, # = Significance PCD vs. HC, + = Significance CF vs. HC.

**Figure 3 jcm-12-06010-f003:**
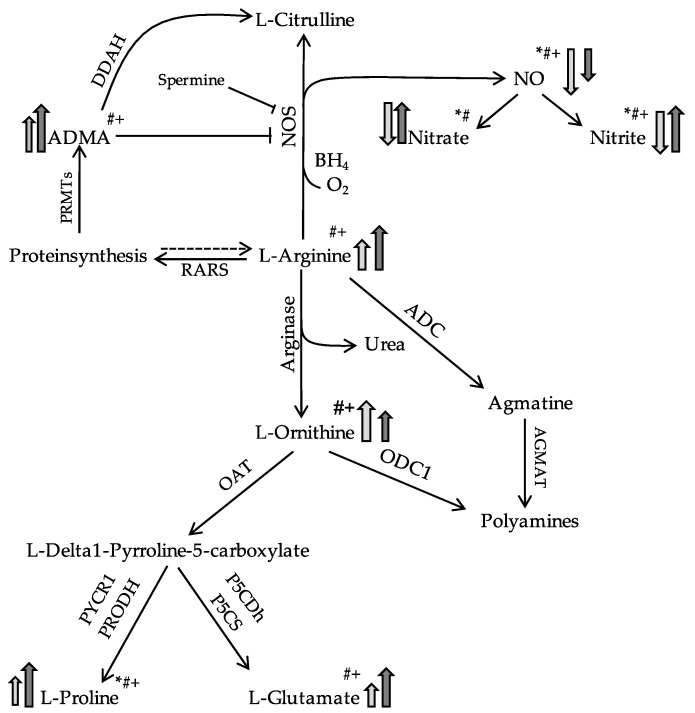
Simplified schematic showing differences in L-Arg/NO pathway in the sputum in PCD (light grey) or CF (dark grey) in comparison to a healthy state. Modified according to Grasemann et al. [30]. Arrows indicate higher or lower. * = Significance PCD vs. CF, # = Significance PCD vs. HC, + = Significance CF vs. HC. Abbreviations. ADMA, asymmetric dimethylarginine; ADC, arginine decarboxylase; AGMAT, agmatinase; BH_4_, tetrahydrobiopterin; DDAH, dimethylarginine dimethylaminohydrolase; NO, nitric oxide; NOS, nitric oxide synthetase; OAT, ornithine aminotransferase; ODC1, ornithine decarboxylase; PRMTs, protein arginine N-methyltransferases; RARS, arginyl-tRNA-synthetase.

**Table 1 jcm-12-06010-t001:** Demographic and clinical data of patients with PCD or CF and of healthy controls (HC). Data are presented as mean ± standard deviation or median (25–75th interquartile range).

	PCD	CF	HC	*p*-Value
Subject (*n*)	24	25	14	
Female (*n* (%))	15 (62.5)	8 (32.0)	8 (57.1)	0.083
Age (years)	16.6 [14.3–21.6]	20.6 [13.7–23.9]	23.7 [17.0–25.5]	0.224
BMI (kg/m^2^)	21.77 ± 4.76	19.88 ± 3.38	21.04 ± 3.25	0.247
BMI (z-Score)	−0.02	−0.49	−0.08	0.149
	[−0.56–0.99]	[−0.98–0.13]	[−1.66–1.57]	
Subjects with:				
Short-term antibiotic therapy < 3 months (*n* (%))	4 (16.7)	8 (32.0)	0 (0)	
Long-term antibiotic therapy (*n* (%))	6 (25)	10 (40)	0 (0)	
CRP > 5 mg/L (*n* (%))	6 (27.3) ^a^	8 (33.3) ^f^	n.m.	0.664
Leucocytes > 10,000/µL (*n* (%))	5 (23.8) ^b^	6 (24.0)	n.m.	0.988
Bacterial colonization (n (%))	**10 (43.5) ^c,^***	**20 (80)**		**0.021**
- *Staphylococcus aureus*	5 (21.7)	12 (48)	n.m.	
- *Pseudomonas aeruginosa*	1 (4.4)	7 (28)		
- *Haemophilus influenzae*	5 (21.7)	0 (0)		
FENO (nL/min)	1.85 ± 1.14 ^d^	4.49 ± 4.24 ^g^	4.88 ± 4.31	0.051
nNO (nL/min)	**10.67 ± 7.82 ^c,^*^,#^**	**96.4 ± 53.73 ^d,+^**	**230.98 ± 53.66**	**<0.001**
FEV1 (z-Score)	−1.88 [−2.59–−1.0]	−1.80 [−2.77–−0.31]	n.m.	0.859
LCI 2.5% (absolute value)	11.36 ± 2.2 ^e^	10.45 ± 5.30 ^g^	n.m.	0.265

Incomplete data available for highlighted variables: ^a^ *n* = 22, ^b^ *n* = 21, ^c^ *n* = 23, ^d^ *n* = 15, ^e^ *n* = 20, ^f^ *n* = 24, ^g^ *n* = 18. Statistics: Bold indicates statistical significance; * = Significance PCD vs. CF; ^#^ = Significance PCD vs. HC; ^+^ = Significance CF vs. HC. Abbreviations: Body Mass Index (BMI); C-reactive Protein (CRP); fractional exhaled nitric oxide (FENO); nasal nitric oxide (nNO); Forced expiratory volume in 1 s (FEV1); Lung Clearance Index (LCI); not measured (n.m.). bold indicates statistical significance.

**Table 2 jcm-12-06010-t002:** Contents in sputum (µmol/mg) of metabolites of the L-Arg/NO pathway in patients with PCD or CF and in healthy controls (HC). Data are presented as median (25–75th interquartile range).

Sputum	PCD	CF	HC	*p*-Value
Subjects (*n*)	24	20	11	
L-Arg	18.43 ^#^ [9.50–44.63]	29.46 ^+^ [18.48–47.22]	9.86 [6.53–14.65]	* 0.067 **^#^ 0.012** **^+^ <0.001**
hArg	0.009 ^#^ [0.005–0.028]	0.037 ^+^ [0.015–0.074]	0.005 [0.003–0.007]	* 0.067 **^#^ 0.019** **^+^ 0.011**
ADMA	0.055 ^#^ [0.029–0.163]	0.015 ^+^ [0.056–0.213]	0.010 [0.008–0.018]	***** 0.687 **^#^ 0.002** **^+^ 0.003**
Nitrite	4.54 *^,#^ [3.35–6.31]	16.28 ^+^ [9.12–20.07]	9.26 [6.98–11.53]	*** <0.001** **^#^ 0.023** **^+^ 0.026**
Nitrate	12.86 *^,#^ [10.27–23.28]	56.83 [31.12–72.06]	40.33 [24.37–47.27]	*** 0.002** **^#^ 0.008** ^+^ 0.09
Orn/Cit	10.20 ^#^ [3.85–57.13]	22.84 ^+^ [11.20–42.76]	1.60 [0.86–3.58]	* 0.884 **^#^ 0.002** **^+^ <0.001**
L-Arg/ADMA	239.33 ^#^ [194.38–406.26]	319.11 ^+^ [154.78–649.33]	917.62 [693.49–1205.32]	* 0.522 **^#^ <0.001** **^+^ <0.001**

Statistics: Bold indicates statistical significance; * = Significance PCD vs. CF; ^#^ = Significance PCD vs. HC; ^+^ = Significance CF vs. HC. Abbreviations. L-Arg, L-Arginine; hArg, Homoarginine, ADMA, asymmetric dimethylarginine; Orn/Cit, Ornithine/Citrulline ratio. bold indicates statistical significance.

## Data Availability

Although most data are unavailable due to ethical restrictions, an enquiry about the research data and analysis can be made to the Department of Pediatric Pneumology, University Children’s Hospital, Ruhr-University Bochum (see correspondence).

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
