# Peer review of "Impaired Nitric Oxide Synthetase Activity in Primary Ciliary Dyskinesia—Data-Driven Hypothesis"

_jcm, 2023, doi:10.3390/jcm12186010_

Round 1
Reviewer 1 Report
The primary aim of this study was to analyze whether the L-Arg/NO pathway is different in PCD, CF and healthy controls (HC). The authors hypothesized that pwPCD and pwCF show higher sputum concentrations of Arg and ADMA in comparison to HC. Yet, authors hypothesized that the Arg/NO pathway behaves differently in PCD and CF, and that these differences account for the PCD-immanent very low nNO levels. ut higher in CF (nitrite 16.28, p < 0.001; nitrate 56.83, p = 0.002).
The results of our study indicate that PCD, unlike CF, is associated with impaired NO synthesis in the lung, presumably due to mechanochemical uncoupling.
This manuscript contains very useful information to understand the low levels of NO in PCD. We knew that this was so but we did not know the biochemical bases. This manuscript provides information in this regard, although it is not conclusive.
Thus, we also know that nasal NO behaves differently in upper and lower airway, in the sense that in many sinonasal inflammatory processes nasal NO is decreased and it is not in patients with PCD. This is so important that the determination of nasal nitric oxide lacks sufficient specificity for the diagnosis of PCD. I think the authors should add some comment about it.
High levels of NO have been reported in some patients with PCD. I would ask the authors to explain if in any of their patients the bronchial NO levels were high and, where appropriate, to correlate it with ciliary mobility, structure and genetics.
It has not been possible to establish correlations between NO levels and certain genetic mutations due to the sample size, a weakness that the authors comment on but that would be of great interest to know for clinical diagnosis.
In table S1 the term CF in its heading appears to be incorrect or misleading.
Reviewer 2 Report
This is a well written manuscript, presenting data from a prospective study, aiming to add information about the L-Arg/NO pathway in PCD patients compared to CF and healthy controls.
The main findings are that the L-arg/NO pathway seems to be altered in a similar way both in CF and PCD and this can possibly be due to airway inflammation. Another main finding is that in PCD and not in healthy controls or CF patients, NO metabolites – nitrate and nitrite were reduced. It is proposed that these results support the mechano-chemical hypothesis to explain low nNO levels in PCD.
To date many different theories aim to explain why nasal NO is low in PCD patients, and the precise mechanism is not clear yet, therefore any added information is significant.
Although the manuscript is interesting and well written, two main points should be clarified –
1. Nasal NO measurements reflect NO in the paranasal sinuses. Studies show that FENO is also somewhat lower in PCD, but does not distinguish PCD patients from others similar to nasal NO measurements. It therefore should be clarified why metabolite levels in sputum necessarily explain alterations in nasal NO measurements.
2. It is postulated that the results support a “mechanic- chemical” decoupling hypothesis. The main problem with this theory as pointed out by Walker et al (DOI: 10.1183/09031936.00176111) is that PCD phenotypes with hyper-frequent or motile dyskinetic cilia have low nitric oxide similar to patients with static cilia. This is also seen in the results quoted in reference #45. Despite the relatively small numbers, it would be interesting to see if there are any differences that can be seen between the 16 patients in this study that were found to have immotile cilia compared to the 5 with dyskinetic cilia.
Specific points:
Introduction:
Line 46-49, suggest referring also to ATS guidelines (although very different), together with ERS guidelines.
Line 70 – correct typing error – Arg residues is proteins…
Methods –
Was the study approved by an ethics board comity – should be noted in the text
Figure 1 – please explain “double inclusion”
Diagnostic parameters for PCD – for PCD patients without genetic confirmation – was genetic testing attempted?
Patients unable to produce sputum were rejected, why wasn’t induced sputum used similarly to the healthy controls?
Results:
Line 185 – correct moderators ->modulators
Line 212 – reference to figure s4, but seems to be s5. Figure axes should have labels.
Table S1 – not clear why “CF” is written in the headline.
Discussion:
Lines 325 -328
This is not clear, NO was significantly lower in upper airways (nasal NO) that is clear, but was it lower in the lower airways? – this seems very subtle and actually did not reach statistic significance (p=0.51). do metabolites in sputum represent correctly what happens in the upper airways?
Lines 351-353
The correlation between nasal NO and ciliary beat frequency (M. Pifferi et al) is interesting, but it should be noted that the correlation was mild – and in the words of the authors – “It has to be acknowledged that the scatter is wide…”.
Line 367 – the reference sighted (46) deals with specific mutations of CFAP74, not sure how generalizable that is.
Line 402-403 –
As correctly stated, most PCD patients have low nNO, that includes those with hypo, hyper and normo- kinetic ciliary beat frequency, thus questioning the “mechanic- chemical” hypothesis
Round 2
Reviewer 2 Report
The two main points raised where answered by the authors in the point by point file, but unfortunately this answers were not implemented in the text.
I would suggest to discuss in the text how sputum analysis is related to nasal NO
I would suggest to discuss the problems with the "mechano-chemical" theory in the text.
